

# Detection of human Metapneumovirus infection in children under 18 years old hospitalized in Lima-Peru

Juana del Valle-Mendoza[1,2,*], Fiorella Orellana-Peralta[1,*], Luis J. del Valle[3], Eduardo Verne[4,5], Claudia Ugarte[4,5], Claudia Weilg[1], Wilmer Silva-Caso[1,2], Jorge Valverde-Ezeta[1], Hugo Carrillo-Ng[2,4], Isaac Peña-Tuesta[1], Carlos Palomares-Reyes[1], Angela Cornejo-Tapia[1] and Miguel Angel Aguilar-Luis[1,2]

[1] School of Medicine, Research and Innovation Centre of the Faculty of Health Sciences., Universidad Peruana de Ciencias Aplicadas, Lima, Peru
[2] Laboratorio de Biologia Molecular, Instituto de Investigación Nutricional (IIN), Lima, Peru
[3] Barcelona Research Center for Multiscale Science and Engineering, Departament d'Enginyeria Química, EEBE, Universitat Politecnica de Catalunya (UPC), Barcelona Tech, Barcelona, Spain
[4] Facultad de Medicina, Universidad Peruana Cayetano Heredia, Lima, Peru
[5] Servicio de Pediatria, Hospital Nacional Cayetano Heredia, Lima, Peru
[*] These authors contributed equally to this work.

Corresponding authors
Juana del Valle-Mendoza,
joana.del.valle@gmail.com,
juana.delvalle@upc.pe
Miguel Angel Aguilar-Luis,
miguel.aguilar@upc.pe,
ma23aguilar@gmail.com

## ABSTRACT

**Background.** Human Metapneumovirus (hMPV) is a negative single-stranded RNA virus. Infection by hMPV mainly affects the pediatric population and can cause upper or lower respiratory tract pathologies which can develop life threating complications. This study was carried out between 2009 and 2010 in a high complexity national hospital in Lima, Peru. The time frame corresponds to the pandemic of influenza A H1N1.

**Methods.** A prospective study was performed between September 2009 and September 2010. Patients with a clinical diagnosis suggestive of an acute respiratory infection were included. RT-PCR was utilized to attain the amplification and identification of the hMPV.

**Results.** A total of 539 samples were analyzed from patients with a clinical context suggestive of an acute respiratory tract infection. Of these samples 73, (13.54%) were positive for hMPV. Out of the positive cases, 63% were under one year old, and increased to nearly 80% when considering children younger than two years old. Cough was the most frequent symptom presented by our population with a number of 62 cases (84.93%). Viral seasonality was also established, noting its predominance during the months of summer in the southern hemisphere. The infection by hMPV has an important prevalence in Peru. It mainly affects children under one year old and should be considered an important differential diagnosis in a patient with an acute respiratory infection.

## INTRODUCTION

The Human Metapneumovirus (hMPV) is a negative single-stranded RNA virus. It belongs to the family Pneumoviridae, which includes the genus Metapneumovirus and Orthopneumovirus, whose main representatives are the Human Metapneumovirus (hMPV) and the Respiratory Syncytial Virus, respectively (*Van den Hoogen et al., 2003*). The hMPV was initially isolated and described in the Netherlands during the year 2001; however, it is presumed that hPMV is responsible for respiratory infections dating back to approximately six decades (*Van den Hoogen et al., 2003*; *Boivin et al., 2002*; *Nissen et al., 2002*; *Peret et al., 2002*; *Boivin et al., 2004*). According to the phylogenetic analysis, within the most frequent isolated genotypes, two subgroups of hPMV have been identified: A and B, which include subtypes A1, A2, B1 and B2 (*Boivin et al., 2004*; *Mackay et al., 2006*).

Infection due to hMPV mainly affects the pediatric population and can cause diverse pathologies of the upper or lower respiratory tract, which include rhinopharyngitis, laryngitis, croup, pneumonia, bronchiolitis or asthma exacerbations (*Williams et al., 2004*; *Boivin et al., 2007*). Furthermore, it also affects the adult population, predominantly elderly and immunosuppressed patients (*Boivin et al., 2007*). Although the clinical context most frequently consists of an asymptomatic period that lasts approximately seven days post exposure, followed by a week of symptoms of an upper respiratory tract infection with gradual resolution, the clinical course can progress toward the lower respiratory tract resulting in lung parenchymal involvement and cause further complications (*Boivin et al., 2003*; *Falsey et al., 2003*; *Mullins et al., 2004*; *Xepapadaki et al., 2004*; *Schlapbach et al., 2011*). It is estimated that infections caused by hMPV are accountable for approximately 20,000 hospitalizations per year in children under five years old in the United States (*Edwards et al., 2013*). However, the available information regarding its epidemiology is limited due to a deficient suspicion and its clinical similarity with other respiratory viruses such as the Influenza virus and the Respiratory Syncytial Virus. It has been described and established in Western countries that the prevalence of the Human Metapneumovirus increases during the late winter months and the beginning of the spring; however, it has also been reported during late spring and summer months in cities such as Hong Kong (*Van den Hoogen et al., 2003*; *Williams et al., 2004*; *Peiris et al., 2003*; *Døllner et al., 2004*; *Haynes et al., 2016*). Currently, the limited availability of molecular methods for the precise diagnosis of viral pathologies such as those caused by hMPV leads to poor epidemiological data in Latin American countries such as Peru. The aim of this study is to identify the Human Metapneumovirus responsible for acute respiratory infections in children from September 2009 to September 2010 in Lima, Peru.

## MATERIALS AND METHODS

### Patients

We performed a prospective study between September 2009 and September 2010 in a high complexity national hospital in Lima, Peru. The time frame corresponds to the pandemic of influenza A H1N1. In this study we included all the patients with clinical diagnosis of acute respiratory infections (ARI). We use the operative definition of ARI,

according to the sanitary directives of the Ministry of Health of Peru, as any infection that involves one or more parts of the respiratory system, with a duration of less than 14 days and with the presence of one or more clinical signs: cough, rhinorrhea, nasal obstruction, odynophagia, dysphonia, otalgia, noisy breathing, respiratory distress, which may or may not be accompanied by fever. The epidemiological and clinical data were: age, clinical symptoms (fever: defined as temperature higher than 38 °C, rhinorrhea, cough, respiratory difficulty, sore throat, wheezing, discomfort, pharyngeal congestion, expectoration, vomiting, diarrhea, others.).

### Ethics statement

Samples were collected following a written informed consent signed by all the parents or respective caregiver because the patients were under 18 years of age. This study has been approved by two independent Ethics Committees from *Hospital Nacional Cayetano Heredia* (Ethical Application N° 021-09) and *Instituto de Investigación Nutricional* (Ethical Application N° 279-2009 / CEI-IIN) in Lima, Peru.

### Samples

To obtain the samples we used two different swabs. A first swab was inserted into both nostrils parallel to the palate to obtain the nasopharyngeal samples (Mini-Tip Culture Direct; Becton-Dickinson Microbiology System, Franklin Lakes, NJ, USA). The second swab was inserted into the posterior pharyngeal and tonsillar areas (Viral Culturette; Becton-Dickinson Microbiology Systems, Franklin Lakes, NJ, USA). A tube containing viral transport medium (minimal essential medium with 2% fetal bovine serum, amphotericin B 20 μg/ml, neomycin 40 μg/ml,) was used to store both swabs. Two aliquots of each fresh specimen were stored at −20 °C to be later analyzed for hMPV.

### RNA extraction

The viral genetic material was extracted used the High Pure RNA Isolation Kit (Roche Applied Science, Mannheim, Germany) to perform the RNA extraction from 200 μL of the samples, according to the manufacturer's instructions. Viral RNA obtained after the extraction was eluted in 100 μl of nuclease-free water.

### Reverse Transcriptase Polymerase chain reaction (RT-PCR) for the analysis of Human Metapneumovirus (hMPV)

A one-step RT-PCR was performed using Real Time ready RNA Virus Master (Roche Diagnostic, Deutschland-Mannheim, Germany) and 250 nM specific primers for hMPV in a final volume of 20 μL. The primers were described by *Van den Hoogen et al. (2003)*, amplifying a conserved fragment of 170 nt in the polymerase chain. Five microliters of the extracted RNA was combined with 15 μl of the master mix and the reverse transcription step was performed 45 °C for 45 min and the amplification consisted of an initial incubation at 95 °C for 2 min, followed by 45 cycles of 95 °C for 1 min; 50 °C for 1 min, and 72 °C for 30 s; with a final extension at 72 °C for 10 min.

Amplified products were recovered from the gel, purified (SpinPrep Gel DNA Kit; San Diego, CA) and sent for commercial sequencing (Macrogen, Seoul, South Korea).

**Table 1  Frequency and prevalence of the hMPV infection.**

| Cases | PCR | | | |
| --- | --- | --- | --- | --- |
| | Frequency (N) | Prevalence (%) [CI 95%] | Odds [CI 95%] | Odds ratio [CI 95%] |
| Positives | 73 | 13.54 [10.91–16.69] | 0.16 [0.11–0.20] | 0.03 [0.01–0.06] |
| Negatives | 466 | 86.46 [83.31–89.09] | 6.38 [5.14–7.92] | |
| Total | 539 | 100.00 | | |

## Statistic analysis

The data was registered in a database designed in Access and were exported to an Excel file for further analysis. The tabulated data were shown as percentage frequencies or counts. The graphics were prepared with the OriginPro v8 software, and analyzed with the Minitab v18 software.

## RESULTS

A total of 539 samples belonging to patients with a clinical context suggestive of an acute respiratory tract infection were analyzed. Of these samples, 73 were positively identified using RT-PCR for hMPV. This finding establishes a prevalence of 13.54% for hMPV in the population studied. Nonetheless, the probability or risk of associating a clinical course of a symptomatic acute respiratory tract infection to the presence of hMPV was less than 3% as demonstrated by the calculated OR (Table 1). This result suggests that other etiological agents (p.e. RSV, influenza, etc.) have a greater probability to be present in the setting of an acute respiratory tract infection (*Appak et al., 2018*; *Brini Khalifa et al., 2018*).

This study also found that 63% of the positive cases for hMPV identified through PCR corresponded to children under one year of age, and this percentage increased to around 80% when considering children under two years of age. The same distribution was observed in the total cases for acute respiratory tract infections. Thus, in both cases, the age distribution can be adjusted to exponential models (Fig. 1A) and where it can clearly be seen that acute respiratory tract infections are more frequent children under two years of age. Additionally, it can be indicated that the sex of children is not a factor associated with the prevalence of hMPV (Fig. 1B).

More than 90% of children diagnosed with an hMPV respiratory tract infection identified positively by PCR were hospitalized. In these patients, the most common signs and symptoms were cough which was present in 84.93% cases ($n = 62$), followed by fever with 75.34% ($n = 55$) and rhinorrhea 71.23% ($n = 52$). In addition, 51 patients (69.86%) presented with dyspnea. This distribution of signs and symptoms was similar for both the population of children with a negative PCR for hMPV, and for all the children clinically diagnosed with an acute respiratory tract infection (Fig. 2A). It is important to note that of the total of cases studied ($n = 539$), mortality was observed in three patients who happened to be negative via PCR for hMPV infection.

Figure 2B, shows the correlation analysis of the signs and symptoms identified in the group of positive patients for hMPV identified through PCR and the total of patients with an acute respiratory tract infection. Also, between positive and negative patients for

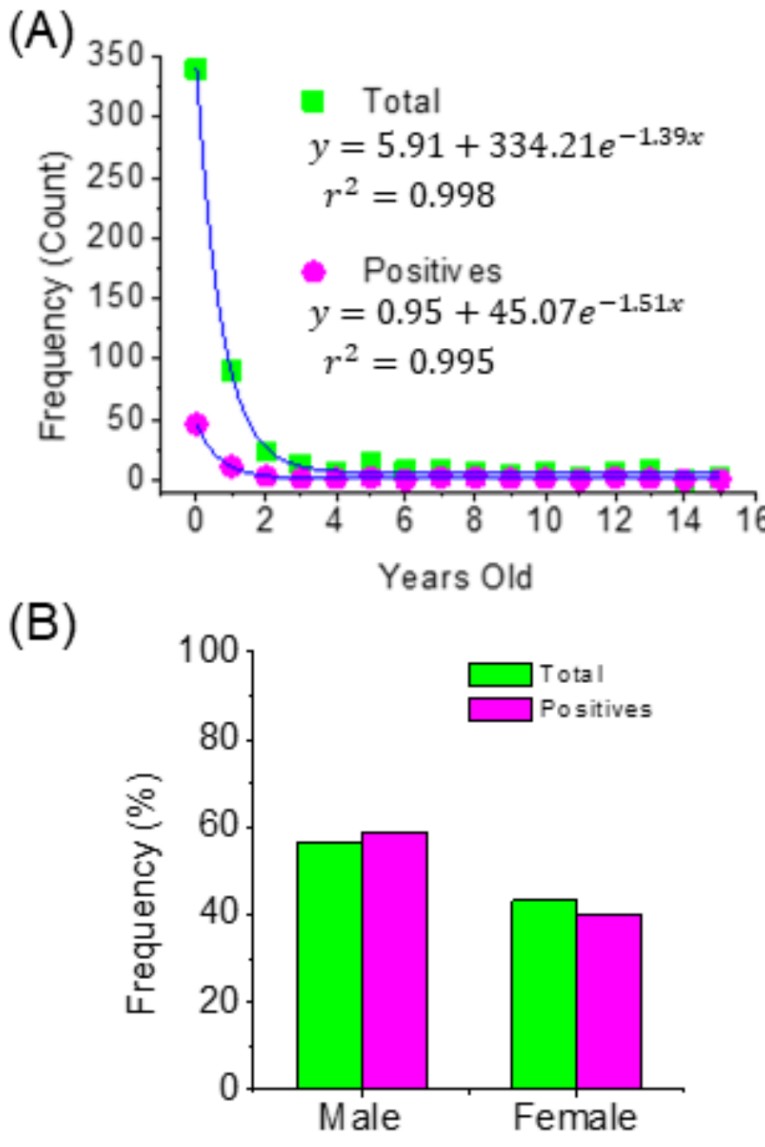

**Figure 1** Distribution of patients by age (A) and sex (B).

hMPV identified through PCR. Clearly, it is observed that in all these cases there is a high correlation ($r \sim 1$). This emphazises that a patient with a respiratory tract infection caused by hMPV, can present a wide range of signs and symptoms which can hardly be discriminated from a clinical diagnosis of an acute respiratory tract infection. Respiratory tract infections caused by hMPV have been described as a seasonal pattern of infectious disease (*Appak et al., 2018*; *Brini Khalifa et al., 2018*). Figure 3 shows the monthly and seasonal distribution of the acute respiratory tract infections cases and those with a positive PCR for hMPV. The cases of hMPV appear in the month of December 2009 ($n = 10$), increased during the month of January 2010 ($n = 15$) and the highest number of positive cases were recorded during the month of March 2010 ($n = 21$) coinciding with the maximum number of cases

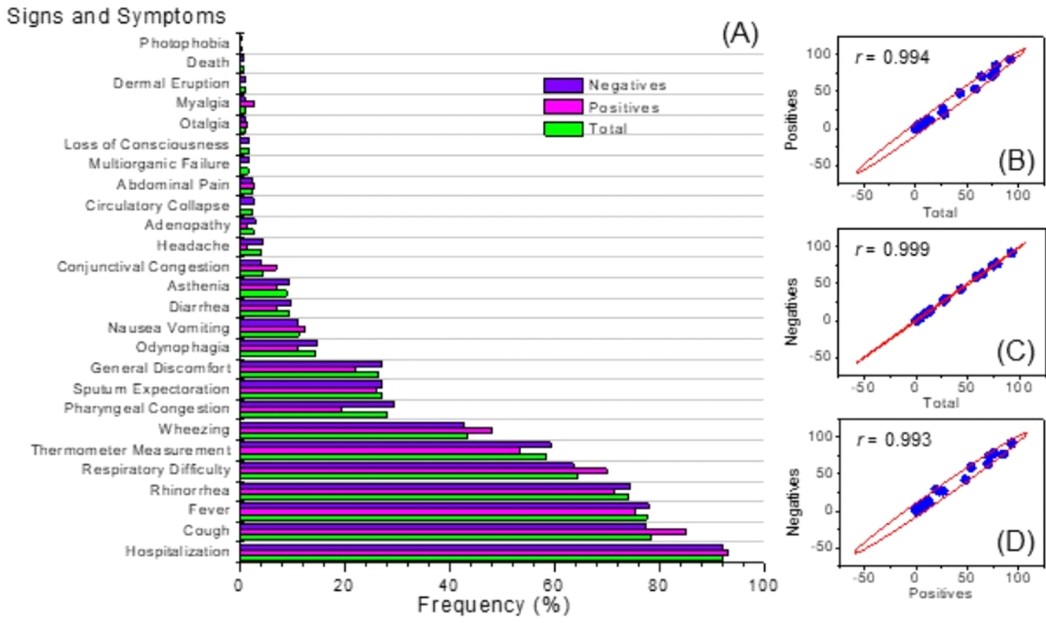

**Figure 2 Clinical symptoms.** Frequency distribution (A) and correlation analysis (B).

diagnosed clinically as an acute respiratory tract infection. Seasonality (in this hemisphere) corresponds to 42 cases during the months of summer, 21 cases were reported in the fall and spring was the season with the least prevalence of cases ($n = 3$) (Fig. 3).

## DISCUSSION

After its first identification and description in 2001, the Human Metapneumovirus (hMPV) has been detected in patients with respiratory tract infections in Europe, Asia, Oceania, Africa, South America (*Hamelin, Abed & Boivin, 2004*; *Druce et al., 2005*). In Peru, between the years 2002 and 2003, genotype B1 was isolated by RT-PCR in mainly patients under five years old (*Gray et al., 2006*). Nonetheless, due to the clinical course similarity it shares with other respiratory viruses it presents as an obstacle for healthcare providers at the time of establishing the etiological diagnosis of the disease. Therefore, its incidence, most affected population, most frequent clinical presentation and seasonal distribution are not well determined. In the present study, 73 cases were isolated during the 13 months in which samples from patients with a clinical course similar to an influenza infection were evaluated. Of the sampled analyzed 13.54% were positive via PCR for hMPV infection, which represents a percentage that merits consideration of hMPV infection as a differential diagnosis in patients with the aforementioned clinical picture. The available information on seroprevalence suggests that the majority of patients have been infected before the age of five (*Van den Hoogen et al., 2003*). Various studies report a prevalence between 5% and 15% in the pediatric population with a clinical course similar to influenza (*Williams et al., 2004*; *Boivin et al., 2003*; *Døllner et al., 2004*; *Esper et al., 2004*; *Louie et al., 2007*; *Schuster et al., 2015*; *Corti et al., 2013*; *Bhattacharyya et al., 2015*; *Williams et al., 2006*). It is also known

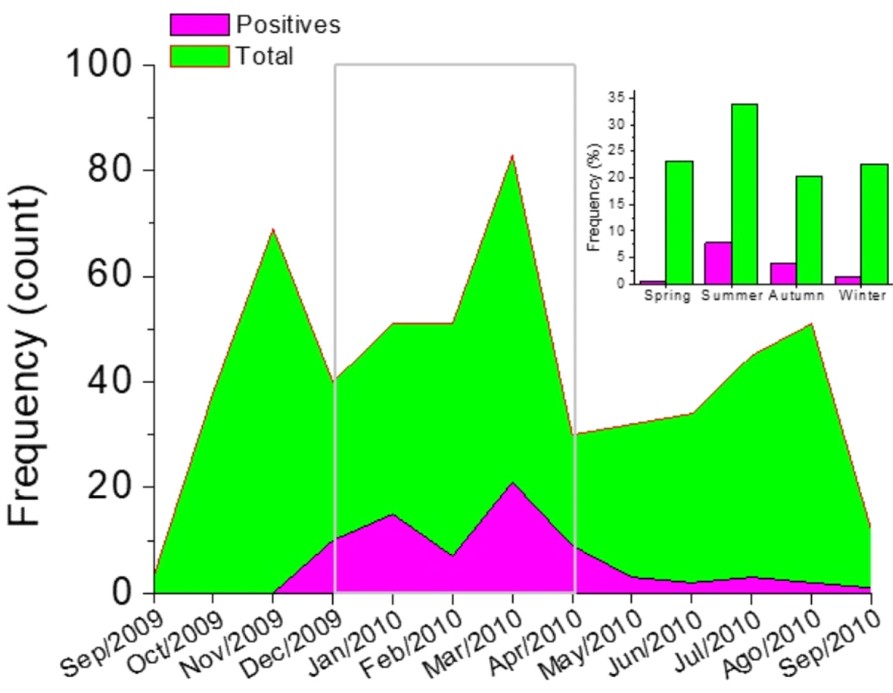

**Figure 3 Monthly distribution of cases.** The gray box indicates the months of greatest incidence. Inset, seasonal distribution.

that the majority of cases correspond to children under one year of age and that early age is associated with greater need for hospitalization (*Edwards et al., 2013*). The findings in the present study are consistent with the literature reviewed, in which 63.01% of positive cases correspond to children under 1 year and 84.93% of cases belong to children under five years old. In relation to the symptoms presented by our study population, (*Williams et al., 2004*; *Edwards et al., 2013*; *Esper et al., 2004*) the most frequent symptom was cough, a finding that coincides with our study, in which this symptom was observed with a frequency of 84.93% in the patients who were positive for hMPV. The second most frequent symptom found in our study was fever (75.34%); however, according to different studies it can present with a prevalence ranging from 52% to 86% depending on the series. On the other hand, dyspnea was identified in 69.86% of the patients, which indicates that the lower respiratory tract is usually compromised. In addition, although the classical theory on viral replication suggests that hMPV infection is limited to the respiratory tract, cases have been reported of viral encephalitis in children with respiratory tract infections by hMPV, in which the virus has been isolated postmortem from the cerebrospinal fluid (CSF) (*Madhi et al., 2006*; *Schildgen et al., 2005*). Of the total number of samples analyzed, four samples came from patients with a clinical diagnosis of Encephalitis; however, after rigorous analysis, none was positive for hMPV infection. It has been described that Human Metapneumovirus infection can cause exacerbations of obstructive conditions such as Asthma and Chronic Obstructive Pulmonary disease (*Williams et al., 2005*; *Vicente et al., 2004*; *Beckham et al., 2005*). This is important in the context of respiratory infections caused by viruses such as RSV that cause

Bronchiolitis in the pediatric population and it has been associated to the development or susceptibility to suffer from asthma in the future (*Openshaw, Dean & Culley, 2003*). In addition, the possibility of partial cross-immunity between both viruses is evaluated, since monoclonal antibodies are capable of neutralizing RSV and hMPV (*Schuster et al., 2015*; *Corti et al., 2013*). The diagnosis of an infection caused by hMPV can be made through serology, direct immunofluorescence, viral culture and RT-PCR. Currently, molecular diagnosis by RT-PCR, either conventional or real-time, offers the highest sensitivity than the rest of diagnostic methods previously described (*Van den Hoogen et al., 2003*; *Williams et al., 2004*; *Boivin et al., 2003*; *Falsey et al., 2003*; *Esper et al., 2004*). In the present study, all the samples were analyzed via RT-PCR to establish the diagnosis of hMPV infection. The seasonality of the virus was observed in the majority of studies during the spring season in western countries; however, in the present study, 57.53% of cases occurred during the summer months, with the month of March exhibiting the highest number of cases ($n = 21$). This finding could be partly explained because more than 80 ARIs were reported during that month, a higher number than any other period during the study. This finding emphasizes the importance of describing the seasonal behavior of hMPV and respiratory viruses that cause similar respiratory clinical courses, so that doctors can use as a tool to make more targeted diagnoses.

A limitation in this study was that we did not evaluate other viral or bacterial etiological agents that could be responsible for coinfections. The clinical characteristics in the presence of coinfection could differ from those with unique infection by hMPV. Another limitation is that the type of sampling in our study supposes a potential selection bias. However, due to our interest in studying the presence of hMPV in patients with ARI, it was the most feasible and efficient sampling selection approach for the context in which it was performed.

## CONCLUSION

Respiratory tract infections caused by hMPV has an important prevalence in Latin American countries, such as Peru. It mainly affects children under one year old and can cause upper and lower respiratory tract infections. It is important to conduct further studies and to provide sustained epidemiological surveillance that allows the attending physician to have the pertinent information available to suspect an infection by hMPV. Moreover, molecular diagnostic methods such as RT-PCR still require standardization; however, it would be ideal if they were available in reference laboratories from all health care facilities in order to provide a specific etiologic diagnosis with the best sensitivity and specificity available.

## ACKNOWLEDGEMENTS

We thank to the staff of the *Hospital Nacional Cayetano Heredia* from Lima, Peru.

### Funding

This work was supported by Incentive for Research of the Universidad Peruana de Ciencias Aplicadas (N°A-067-2019-UPC), Lima, Peru. The funders had no role in study design, data collection and analysis, decision to publish, or preparation of the manuscript.

### Grant Disclosures

The following grant information was disclosed by the authors:
Incentive for Research of the Universidad Peruana de Ciencias Aplicadas: N°A-067-2019-UPC.

### Competing Interests

The authors declare there are no competing interests.

### Author Contributions

- Juana del Valle-Mendoza conceived and designed the experiments, analyzed the data, contributed reagents/materials/analysis tools, authored or reviewed drafts of the paper, approved the final draft.
- Fiorella Orellana-Peralta performed the experiments, approved the final draft.
- Luis J. del Valle conceived and designed the experiments, analyzed the data, contributed reagents/materials/analysis tools, prepared figures and/or tables, authored or reviewed drafts of the paper, approved the final draft.
- Eduardo Verne and Claudia Ugarte conceived and designed the experiments, approved the final draft, was responsible for the clinical assessment, samples collection and database completion.
- Claudia Weilg analyzed the data, authored or reviewed drafts of the paper, approved the final draft.
- Wilmer Silva-Caso authored or reviewed drafts of the paper, approved the final draft, was responsible for the clinical assessment, samples collection and database completion.
- Jorge Valverde-Ezeta analyzed the data, prepared figures and/or tables, authored or reviewed drafts of the paper, approved the final draft.
- Hugo Carrillo-Ng performed the experiments, analyzed the data, prepared figures and/or tables, authored or reviewed drafts of the paper, approved the final draft, was responsible for the clinical assessment, samples collection and database completion.
- Isaac Peña-Tuesta performed the experiments, prepared figures and/or tables, approved the final draft.
- Carlos Palomares-Reyes performed the experiments, analyzed the data, contributed reagents/materials/analysis tools, prepared figures and/or tables, approved the final draft.
- Angela Cornejo-Tapia performed the experiments, contributed reagents/materials/analysis tools, approved the final draft.
- Miguel Angel Aguilar-Luis conceived and designed the experiments, approved the final draft.

## Human Ethics

The following information was supplied relating to ethical approvals (i.e., approving body and any reference numbers):

This study has been approved by two independent Ethics Committees from Hospital Nacional Cayetano Heredia (Ethical Application N°021-09) and Instituto de Investigación Nutricional (Ethical Application N°279-2009/CEI-IIN) in Lima, Peru.

## Data Availability

The abstraction format used in the study and the raw data are available at Figshare: https://figshare.com/articles/Dataset_MPVh_y2019m02/7670519.

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
