# Peer review of "Detection of human Metapneumovirus infection in children under 18 years old hospitalized in Lima-Peru"

_PeerJ, doi:10.7717/peerj.7266_

## Round 0.1 · original submission · Major Revisions

Please carefully consider the comments and suggestions provided by the reviewers of your manuscript, and use them to help prepare a revised version of your manuscript.

Thank-you

Reviewer 1 ·

Basic reporting

A few style points in regards to the language.
1) Numbers 1-10 should be spelled out, eg one, two, etc. [lines 36, 37, 40, 63, 68, 143, 144, 147, 156, 178, 187, 190, 192, 193, 204, 227]

2) Sentence starting "Therefore..." on line 180 is unclear, I would clarify

3) Sentence starting "As detailed..." on line 157 is unclear as well, please clarify

In regards to literature references, fix citation on line 74.

Experimental design

It is unclear what kind of study is being described. Is it prospective? I would suggest describing the study type in the abstract under methods and line 83 in the main manuscript.

Validity of the findings

Unable to truly assess without knowing what kind of study. I assume that it is prospective given consent was taken. If prospective, it appears to be statistically sound and conclusion is well stated.

Additional comments

I would add a paragraph describing the limitations to your study and how you addressed those limitations if it all possible.

Although there is great content in the discussion, it can be organized to be made clearer and easier to read. I suggest organizing it as such: paragraph 1 describe what you found in your study, paragraph 2 (or more) describe how your findings compare to what is in the literature, next paragraph your strengths and limitations and last the conclusion.

Reviewer 2 ·

Basic reporting

The language needs to be improved, for example, line 180-182, it does not make sense. The background is well stated. The article is generally well described, including figures and tables.

Experimental design

1. Did you look into the co-infection? Does it differ in characteristics / clinical symptoms / signs from those with single hMPV infection? This might affect your findings particularly considering that the study period overlaps with pandemic influenza A H1N1.
2. The methods in abstract is too short.
3. Line 41: "hMPV can cause both upper and lower respiratory tract infections that can develop into potential life threating complications". Where is this sentence concluded from? You did not mention how many patients were AURI or ALRI in the result. You also did not mention any "life threating complications" among hMPV patients.
4. Did you include all ARI patients during the 2-year study period or did you use any sampling approach?
5. Line 86-88: How did you get the ARI definition? Did you use the same case definition for all patients under 18? Is it appropriate to do so? If you did not include fever as one of the criteria to define ARI, will the result be different (any sensitivity analysis)?
6. Line 93: The underlying population is children under 18 years old. It does not make sense to say "or by the parents or respective guardians in the case of underage patients below 18 years of age".

Validity of the findings

7. I don't see the point of Table 1 (line 138-141). What is your purpose? Also, I don't think it indicates "other etiological agents have a greater association". I guess you wanted to mean "percentage".
8. Line 144: The high percentage of hMPV in infants does not necessarily mean that this is the age group with high risk. The fact that how many ARI patients were tested (denominator) also affects the number of hMPV being positive. This is particularly a problem considering that the tested ARI cases mainly came from infants too. Anyway, can you please clarify the selection bias here?
9. Line 148: "sex of children is not a factor associated with the prevalence of hMPV" conflicts with line 193 "the majority of patients with hMPV infection were male (58.90%)".
10. Line 149-150: It does not make sense "...due to the high frequency of signs and symptoms...". What were you trying to say here? Also, the number of patients who have been hospitalised should be described at the beginning of the result.
11. Line 168: Can you please clarify that there is no selection bias here (the high number of hMPV in March might be due to the large number of ARI samples tested in March)?
12. Line 190: I don't think you have any result indicating this point "early age is associated with greater need for hospitalisation".

---

## Round 0.2 · accepted · Accept

We are happy to accept your manuscript for publication. We just need you to correct the typos identified by reviewer 1, which are minor enough that they can be resolved while in Production.

Reviewer 1 ·

Basic reporting

The language and read of this article has vastly improved. Small typos should be corrected. Line 30, "life threating..." should be corrected to "life threatening...". Line 61, the I in "Furthermore, It..." should be lowercase. Line 169, "emphazises" should be corrected to "emphasizes". And line 192, "Of the sampled..." should be corrected to "Among the samples..." After those few changes, this article is well written and publishable.

Experimental design

The design is stated much more clearly. I have nothing to add.

Validity of the findings

No comment.

Additional comments

Great job on the revisions.